# Temperature Increase During Flexible Ureteroscopic Approach with Holmium Laser Lithotripsy: How Much Should We Be Concerned? [note 1]

**DOI:** 10.3390/medicina61081335

**Published:** 2025-07-24

**Authors:** Razvan Multescu, Petrisor Geavlete, Dragos Georgescu, Cristian Surcel, Catalin Bulai, Cristian Mares, Laurian Maxim, Bogdan Geavlete

**Affiliations:** 1Department of Urology, “Saint John” Emergency Clinical Hospital, 042122 Bucharest, Romania; razvan.multescu@umfcd.ro (R.M.);; 2Faculty of General Medicine, “Carol Davila” University of Medicine and Pharmacy, 020021 Bucharest, Romania; 3Department of Urology, Fundeni Clinical Institute, 022328 Bucharest, Romania; 4Department of Urology, Emergency Clinical County Hospital Brasov, 500326 Brasov, Romania

**Keywords:** holmium laser lithotripsy, temperature variation, irrigation fluid, ureteral access sheath

## Abstract

*Background and Objectives*: The aim of our study was to evaluate in an ex vivo setting the impact of the holmium laser lithotripsy over the temperature of the irrigation fluid. *Materials and Methods*: We recorded temperature changes in an ex vivo porcine model during laser activation using dusting (18 Hz, 0.6 J, 10.8 W) and fragmenting settings (8 Hz, 2 J, 16 W). The temperature was recorded for each of these modes in three settings: without irrigation or access sheath, with irrigation but no access sheath, and with irrigation and a 10/12 F access sheath in place. *Results*: Using dusting settings, the maximum recorded temperatures were 42.3 degrees Celsius (no irrigation, no sheath), 37.3 degrees Celsius (with irrigation but no access sheath) and 36.2 degrees Celsius (with irrigation and access sheath). In fragmenting mode, the maximum recorded temperatures were 52 degrees Celsius (no irrigation, no sheath), 43.1 degrees Celsius (with irrigation but no access sheath), and 42.9 degrees Celsius (with irrigation and access sheath). *Conclusions*: In certain conditions (no irrigation, more watts) the temperature may rise to dangerous levels. However, in closer to real-life settings (with irrigation and especially when ureteral access sheaths are employed) the magnitude of this effect is limited, making flexible intrarenal laser lithotripsy a reasonably safe procedure.

## 1. Introduction

After more than three decades of evolution, a flexible ureteroscopic approach has become a common diagnostic and treatment procedure dedicated to upper urinary tract pathology.

One of the main goals of the endourologists is to treat stone volumes as large as possible, using the least invasive methods available. This involves using various laser settings in order to improve their efficiency but frequently may result in extended laser durations, sometimes with unwanted and potentially dangerous consequences.

The thermal effect generated during laser activation—especially at higher power settings or with prolonged firing—has raised increasing concern regarding its potential to injure renal tissues, and it thus has become a popular study topic.

In the last years, numerous experimental studies tried to demonstrate that irrigation parameters, laser settings, and anatomical constraints significantly influence temperature dynamics within the pyelocaliceal system [1,2]. A critical threshold of 43 °C is often cited as a marker of potential thermal injury, though the precise biological relevance of this value remains debated [3,4]. In clinical settings, irrigation and the use of ureteral access sheaths may mitigate thermal rise, yet real-time temperature monitoring is rarely performed, and safety assumptions are largely extrapolated from preclinical data.

The aim of our study was to evaluate, in an ex vivo setting (but simulating as closely as possible the real-life conditions), the impact of the most commonly used holmium: YAG laser settings on the temperature change of the irrigation fluid. It also aims to investigate whether the utilization of the ureteral access sheath may have an influence on this parameter as well.

To our knowledge, this is one of the few studies to systematically evaluate temperature evolution under commonly used holmium laser settings in a structured ex vivo model, while directly comparing the impact of irrigation and ureteral access sheath use. The strength of our work is that it measures this parameter on a biological model, simulating a plausible clinical scenario, in a controlled and standardized manner. This approach aims to provide practical insights relevant to real-life endourological procedures.

## 2. Materials and Methods

The in vitro setting included the use of an Olympus URF-V3 flexible ureteroscope (Olympus Corporation, Hamburg, Germany), a Dornier Medilas H Solvo 35 Holmium:YAG laser (Dornier MedTech, Wessling, Germany) with a 270 micron laser fiber, a thermometer with metallic probe for intracavitary temperature measurements, and a 10/12 F Innovex ureteral access sheath (Innovex Medical Co., Shanghai, China).

For the measurements we tried to replicate the in vivo conditions using a freshly harvested porcine urinary system. The animal organs were used within 3 h post-harvest, being maintained at ambient conditions; no measures of active heating were applied.

Temperature measurements were performed using a TP-300 digital thermometer (Ningbo Keitai Electronic Co., Ltd., Ningbo, China) equipped with a K-type thermocouple probe, with an operating range of −50 °C to +300 °C and a resolution of 0.1 °C.

The temperature probe was inserted through a puncture in the renal parenchyma into one of the secondary middle calyces (Figure 1), with its tip adjacent to the renal papilla (Figure 2). Initially, we evaluated the situation in which the flexible ureteroscope was inserted directly through the ureter and, after that, the one using a ureteral access sheath.

Once the tip of the ureteroscope was in the same calyx as the temperature probe, the laser was continuously activated for 5 min using a dusting setting (18 Hz, 0.6 J, 10.8 W) and again with a fragmenting setting, respectively (8 Hz, 2 J, 16 W).

In an attempt to replicate real-life intraoperative conditions the fiber-to-thermometer distance was not standardized. The laser fiber was manually maintained at an approximately constant distance of about 2 mm from the thermometer tip under direct vision.

The temperatures were recorded without any irrigation or access sheath and then again with irrigation turned on. For the third set of recordings, a 10/12 F ureteral access sheath was inserted through the ureter and temperatures were measured again with the irrigation turned on, with both laser settings. All the measurements were performed in triplicates, and we calculated the mean values and the standard deviation (SD).

When the ureteral access sheath was used, it was placed with the distal end 1 cm below the uretero-pelvic junction, making sure that the fluid back-flow was evident through the sheath.

Temperature measurement was performed for the entire period of time when the laser was fired, and also for 1 min after the laser activation ended.

For irrigation we used room temperature (20.2 degrees Celsius) saline in 1 liter bags. We used hydrostatic pressure, with the saline bags being suspended at 100 cm above the kidneys, providing a flow of approximately 35 mL/min.

Intra-pyelocaliceal pressure was not directly measured.

One-way analysis of variance (ANOVA) was performed at each time point to assess statistically significant differences in temperature between the three experimental scenarios (no irrigation, irrigation only, and irrigation with ureteral access sheath). A *p*-value lower than 0.05 was considered statistically significant. Data were analyzed using the sets of three measurements for each condition and time point. When ANOVA indicated statistical significance, post hoc evaluation was performed. For the statistical analysis we used Python v3.11 software.

## 3. Results

No macroscopic alterations of the mucosa of the working calyx were visible at the end of each laser activation period.

When the laser was programmed for dusting mode (Table 1, Figure 3), without irrigation fluid or ureteral access sheath, the temperature rose to the maximum of 42.3 degrees Celsius. This maximum value was reached at 1 min of laser activation. With the irrigation turned on, the temperature in the working calyx only rose up to 37.3 degrees Celsius, with the maximum value being reached after 5 min of laser activation. When the ureteroscope was inserted through the ureteral access sheath, the maximum temperature rise was even lower, only up to 36.2 degrees Celsius, in the second minute of laser activation.

ANOVA testing revealed statistically significant differences in temperature between the three experimental conditions at all recorded time points. Post hoc analysis showed that the highest temperatures were consistently recorded in the “no irrigation, no ureteral access sheath” scenario. The addition of irrigation significantly reduced the temperature, while the combination of irrigation and an ureteral access sheath provided the most consistent cooling effect, with statistical significance between minutes two and five.

For dusting settings, with the irrigation on and without an ureteral access sheath, the temperature decreased to 27.1 degrees Celsius in the 20 s after the laser stopped and to 20.2 degrees Celsius after 1 min. With irrigation and when the ureteral access sheath was used the caliceal temperature was 24.9 degrees Celsius at 20 s after the laser deactivation and 20.2 at 1 min.

Using the frequency and energy for the fragmenting effect (Table 2, Figure 4), without irrigation or an ureteral access sheath, a maximum 52 degrees Celsius temperature was reached after 5 min of laser activation. With irrigation, a maximum temperature of 43.1 degrees Celsius was recorded after the first minute of laser activation (only a slight increase over the threshold of 43 degrees Celsius). For the next 4 min, only temperatures lower than 43 degrees Celsius were recorded. When the ureteral access sheath was employed, the temperature rise was limited at a maximum of 42.9 degrees Celsius after the first minute and lower than 42 degrees Celsius in the last 3 min of laser activation.

Similar to the dusting mode, one-way ANOVA applied to fragmenting settings demonstrated statistically significant temperature differences among the four experimental scenarios at all measured time points.

Post hoc analysis confirmed a clear separation between the “no irrigation, no ureteral access sheath” condition and all others. No significant difference was observed between the “irrigation only” and “irrigation and ureteral access sheath” groups during the initial 2 min, suggesting comparable early cooling effects. Statistically significant divergence became more apparent later, by minutes three and four, but by minute five the temperatures in the “irrigation only” and “irrigation and ureteral access sheath” groups converged again.

In this last setting, in 20 s after the laser activation ended, the recorded temperature in the calyx decreased to 31.2 degrees Celsius and after 1 min became 20.3 degrees Celsius. With irrigation but without a ureteral access sheath the caliceal temperature at 20 s after laser deactivation was 36.9 degrees Celsius and 23.3 degrees Celsius at 1 min.

## 4. Discussion

In our study we tried to replicate conditions similar to those encountered during in vivo surgery. It is generally accepted that tissue exposure to increased temperatures for a certain number of minutes may create damage by thermo-coagulative necrosis.

Although the no-irrigation condition does not reflect typical operative endourological practice, it was included as a reference benchmark for comparison, and is consistent with the prior experimental literature. This scenario allowed us to better illustrate the protective effects of irrigation and access sheath use in mitigating temperature rise during laser activation.

The choice of a 5 min continuous laser activation period was intended to simulate a maximal yet plausible clinical scenario, allowing a standardized comparison across various scenarios. Continuous activation is not very common in real-life practice, and may be dangerous. In this regard, some authors reported increased morbidity associated with this style of laser lithotripsy, especially for high-power machines [5]. The current approach enabled us to observe temperature evolution under consistent and controlled circumstances, rather than to determine absolute peak temperatures under prolonged exposure.

In an article published in 1984, Saparato and Dewey described a formula that tried to quantify these exposures, using a threshold temperature of 43 degrees. Above this threshold, the time to coagulate tissue halves with each degree: 120 min for 44 degrees, 60 min for 45 degrees and so on. It is important to note that the authors state that the 43 degree temperature was an arbitrarily chosen value [3]. Although the initial paper’s topic was cancer therapy, almost all papers studying the influence of laser activation over the irrigation fluid report these values.

However, other studies suggested that even for lower temperatures, impairment of renal function may occur [6]. Naturally, there is the complex issue of not only the degree of elevation in the maximum temperature but also the duration for which it is sustained on the tissues.

The dusting technique in larger stone volumes offers significant advantages such as shorter operating time and reduced mechanic trauma of the ureteral wall. Also, it may decrease operating costs by reducing the necessity for more accessory instruments such as baskets for stone fragment retrieval. However, there are some concerns regarding the fact that there is no real “stone free” status after dusting, and that the technique may have a greater recurrence rate by regrowth of the residual stone fragments, thus canceling the initial savings [7,8].

In vivo, long-time dusting sessions are usually necessary when treating large stones, and these procedures are most often, if not always, performed using irrigation and frequently employing ureteral access sheaths [9,10]. Therefore, it is preferable to evaluate the consequences of laser lithotripsy under these settings.

In our study, when the recordings were performed in such conditions, the temperature increase did not reach alarming levels. In this regard, exposure to the recorded temperatures should exceed a rather long period of time in order to have the potential to harm tissues, suggesting that, under the studied conditions, temperature elevations are limited and unlikely to reach levels typically associated with thermal injury. However, in the absence of in vivo validation or histopathological assessment, no direct conclusions can be drawn regarding tissue damage. Also, it should be taken into consideration that most of the urologists limit the intrarenal time of surgery to a certain duration in order to prevent complications. Most of the studies correlate long operating time especially, but not exclusively, with septic complication [11]. Studies found that the lowest complication rate was associated with procedures less than one hour long (accepting that this conclusion may be biased towards simpler procedures). However, many authors recommend limiting the operating time to 90 min [12], less than the 120 min required for the 43 degrees threshold to become significant. Of course, this strategy may imply the necessity of multiple procedures.

The temperature increases further when higher power levels are used, but the phenomenon seems to be limited by the irrigation and ureteral access sheath use. However, such laser settings are probably not employed in vivo for long intervals of time, being applied in short, repetitive intervals of laser activation [10]. In this last regard, another conclusion suggested by our measurements is that the pyelocaliceal system cools rather quickly after the laser activation ends, limiting the thermal effect over the tissues.

We evaluated if the modality to apply the laser energy may influence the temperature variation. The authors compared MOSES technology vs. conventional pulse, finding no significant differences in the increase of temperature between the two settings, using different irrigation types and power settings. Manual pump-assisted irrigation prevented critical temperature rise, even when high power was used (up to 60 W) [13].

There are other factors which should be taken into consideration when trying to quantify temperature impact over the tissues.

A percentage of the temperature can be absorbed by the fragmented stone. Conversely, when using some lithotripsy modes, such as pop-dusting (achieving dusting using “pop-corn” effect, with short laser bursts of higher energy), the major heating effect may be absorbed by the irrigation fluid [14].

Predictably, the heating effect may be influenced by the volume of fluid in which the laser is activated, in other words by the configuration of the pyelocaliceal system. Rezakahn Khajeh et al. found that when the laser fiber was activated at 0.5 J, 80 Hz, 40 W in a space filled with 0.5 mL of fluid, the dangerous temperature was reached in 3 s, if there was no irrigation. In the larger space, filled with 60.8 mL of fluid, the critical temperature was never reached, no matter the type of irrigation employed. Furthermore, irrespective of the volume, the temperature stayed within a safe range at an irrigation flow rate of 40 mL/min [1]. Such risks should be also taken into consideration when laser activation is performed in the narrow space of the ureter.

In our study, when proper circulation of the irrigation fluid was ensured (irrigation was turned on, and especially when a ureteral access sheath was in place, with fluid back-flow evident through it), the temperature increase was significantly lower compared to the no irrigation scenario, thus ensuring a safer environment. However, fluid circulation may sometimes push the heated liquid to the surrounding tissues faster than natural convection [14].

The matter is further complicated by other additional factors. In addition to not being able to thoroughly quantify the impact of laser lithotripsy and other parameters such as irrigation over the temperature on the pyelocaliceal system, we also do not know the exact long-term consequences of these type of histological alterations.

The kidney parenchyma is characterized by high blood flow, which may have a protecting effect in the occurrence (or at least the extent) of thermal damage [14].

An additional layer of complexity arises when taking the healing process into consideration, with some of the lesions being only temporary. Peteinaris et al. studied the lesions that may occur in the porcine kidneys exposed to temperatures higher than 43 degrees Celsius for one hour of lithotripsy using TFL. Histopathological analysis demonstrated severe alterations one week after the procedure but only mild changes at two weeks after it, suggesting that the healing process is capable of improving severe lesions in a short time frame and that exposure even above the 43 degrees threshold may be reasonably tolerated [4].

The use of an ureteral access sheath may have a beneficial influence over the intrarenal temperature rise during laser activation. A paper by Okhunov et al. used an in vivo porcine model to evaluate the influence of dusting, low- and high-power laser lithotripsy over the intrarenal temperature, studying also the impact of the ureteral access sheath use. Without an ureteral access sheath, the critical threshold temperature of 44 degrees Celsius was reached in almost all measurements. By comparison, when a 14 F ureteral access sheath was used, in half of the measurement groups this temperature was not exceeded [15]. In our study, we also observed that the use of an ureteral access sheath for the pyelocaliceal approach resulted in lower recorded temperatures.

For dusting settings, the temperature curves of the “irrigation only” and “irrigation and ureteral access sheath” groups remained distinct, (the differences being statistically significant in post hoc analysis) emphasizing the additive cooling benefit provided by the access sheath even under continuous irrigation.

Post hoc analysis for fragmenting measurements for the “irrigation only” and “irrigation and ureteral access sheath” groups demonstrated the lowest temperature rise from all three scenarios. The differences between those two become statistically significant at minute three and four, but by the end of the study the curves converged again. While this may suggest a late stabilizing trend such as reaching thermal equilibrium, we acknowledge that minor experimental variations—such as fiber-to-thermometer distance—could influence this outcome. This possibility has been noted as a limitation of the study. Other possibly involved variables may be related to the circulation of the irrigation fluid and probably some other factors yet to be determined.

We believe that the recorded temperature differences between experimental conditions may be clinically relevant when considering thermal safety during laser lithotripsy. As noted, while temperatures above 43 °C are often cited as potentially harmful, tissue injury is not determined by a single threshold but rather by the combination of peak temperature and duration of exposure. Thus, reducing intrarenal temperature even by a few degrees, may lower the risk of cumulative thermal injury during prolonged activation periods. While we did not calculate formal effect size metrics, the significant ANOVA results across multiple settings suggest that experimental conditions (and especially irrigation and ureteral access sheath use) may have a substantial impact on temperature outcomes.

An interesting finding of the article cited above was that the temperature measured near the flexible ureteroscope may be significantly different (both higher and lower) than the actual pyelocaliceal temperature [15]. This observation emphasizes the complex endeavor of attempting to determine the actual heat exposure of renal tissues during laser lithotripsy. It introduces the possibility of additional sources of inaccuracy, including the measurement of temperature.

A notable finding in our study is the rather rapid temperature decline observed after laser deactivation, particularly under conditions with active irrigation. This sharp decrease highlights the role of irrigation not only in mitigating temperature rise during lithotripsy but also in facilitating fast thermal recovery once energy delivery ceases. Although not a primary endpoint, this observation reinforces the clinical benefit of using continuous irrigation and intermittent laser activation to limit tissue exposure to prolonged elevated temperatures.

In the current study we only evaluated a 35 W Holmium laser. Further studies involving high-power holmium lasers, TFL and thulium: YAG are required in order to further understand the influence of various types of laser lithotripsy over the temperature in the pyelocaliceal system.

Aldoukhi et al., evaluating, on an in vivo porcine model, the temperature variations during laser activation at 40 W (0.5 J, 80 Hz) found that dangerous temperature was exceeded with no or low-flow irrigation, but was never reached when high-flow irrigation was applied [2].

It was suspected that thulium lasers may generate a greater amount of heat. Belle et al. compared Olympus 60W TFL with two models of low- and high-power holmium: YAG lasers (Dornier 30 W and Olympus 100 W) and phantom stone lithotripsy was performed at power settings ranging from 3.6 to 30 W. TFL generated more heat than holmium machines at all lithotripsy settings, exceeding the 43 degrees Celsius threshold at 30 W [16].

Molina et al. found that the temperature rise was greater for super-pulsed TFL than holmium on the fragmenting setting, without significant differences between the two types on dusting settings. However, it is worth noting that neither setting reached the temperatures associated with thermal injury risk, and no tissue damage was documented at the histological exam [17].

Another study by Ortner et al. comparatively evaluating thulium: YAG, TFL, and holmium: YAG concluded that all lasers create similar temperature changes and have safe temperature profiles at powers lower than 40 W [18].

Æsøy et al., comparing holmium: YAG and TFL lasers, found that temperature increase during laser activation is dependent on the power (more watts, higher the temperature increase), but also on the fiber size. In this regard, the temperature rise was higher when using the holmium laser with a 270 micron fiber than for the TFL with 150- and 200-micron fibers. The authors monitored both intra-cavitary and intra-parenchymatous temperature, finding that pyelocaliceal temperature is influenced by the watts during laser activation. For higher powers (20 and 30 W) and when using large fibers, significant changes of temperature in the renal parenchyma were also recorded [19].

### Study Limitations

This study has several limitations that should be acknowledged.

First, it was conducted using an ex vivo porcine model, which does not fully replicate the physiological conditions of a perfused, living system. In particular, the absence of blood flow, immune response, and thermoregulatory mechanisms limits the extrapolation of our findings to in vivo settings.

A further limitation is the absence of histopathological assessment, which was not pursued given the ex vivo nature of the model. Without vascular perfusion and tissue viability, such analysis would not yield meaningful insights into thermal injury, and definitive conclusions about tissue-level effects cannot be drawn. Therefore, our findings are restricted to thermal dynamics, and any biological implications remain speculative. While in vivo models provide greater realism, the ex vivo setup allowed controlled conditions with minimal variability.

Furthermore, the temperature of the model was not maintained at core body levels, which may limit the direct extrapolation of the results to in vivo conditions. These factors were necessary trade-offs to allow for controlled, reproducible comparisons between different irrigation and access sheath scenarios.

Certain parameters—such as intra-pyelocaliceal pressure and exact fiber-to-thermometer distances—were not directly measured or standardized, and irrigation flow was based on gravity-driven passive flow without active flow control. While these factors introduce some experimental variability, they were intentionally preserved to reflect the variability and realism of clinical practice. Our goal was not to create a fully mechanized setup, but rather to simulate operative scenarios in a reproducible yet pragmatically relevant manner.

Finally, the continuous 5 min laser activation scenario, while chosen to ensure reproducibility and observe cumulative thermal effects, may not fully reflect the intermittent laser usage patterns typically observed in clinical practice.

## 5. Conclusions

During laser lithotripsy, in certain conditions (no irrigation, higher power settings) temperature may increase to dangerous levels for potential tissue damage.

Our results suggest that, in most real-life settings, laser lithotripsy can be performed safely when appropriate thermal protective measures (such as continuous irrigation and laser activation in time-limited bursts) are applied to prevent excessive intrarenal temperature rise. The employment of ureteral access sheaths has the capacity to further reduce the increase in temperature during laser activation, thereby having the potential to prevent or at least restrict tissue injury.

Further studies are required using different types of lasers which are already in clinical use today.

## Figures and Tables

**Figure 1 medicina-61-01335-f001:**
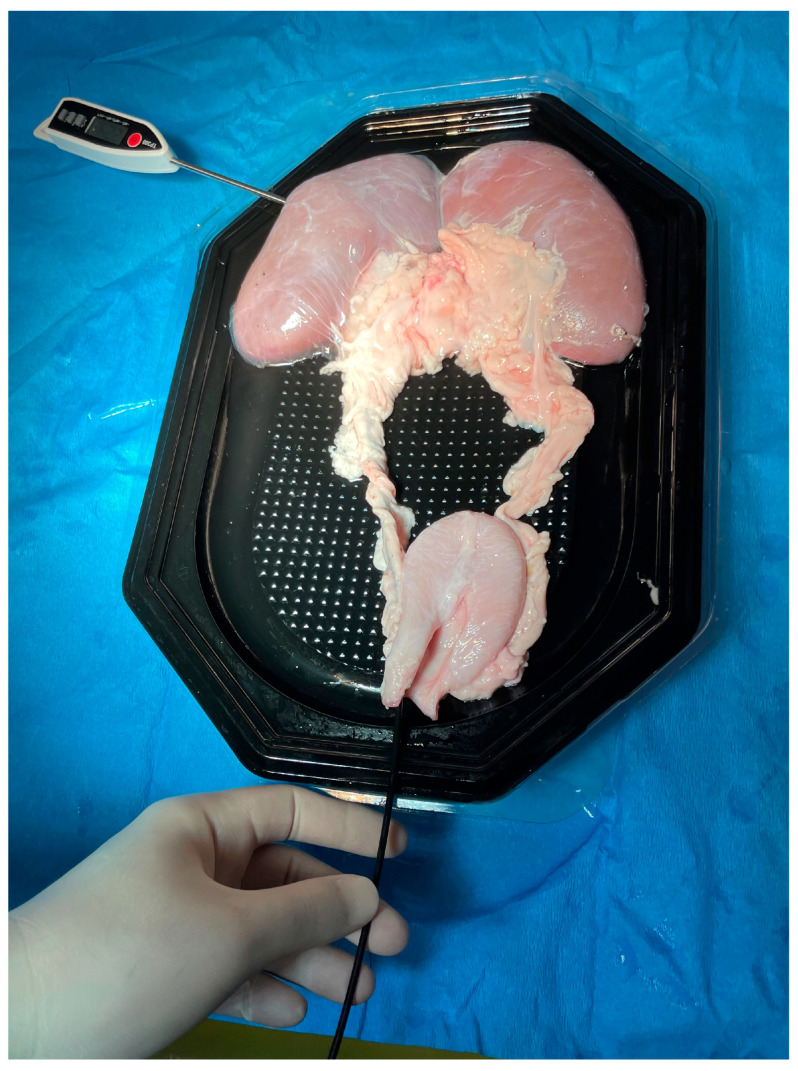
The ex vivo setting for temperature measurements, including the porcine urinary tract with a thermometer inserted through a puncture in the middle calyx.

**Figure 2 medicina-61-01335-f002:**
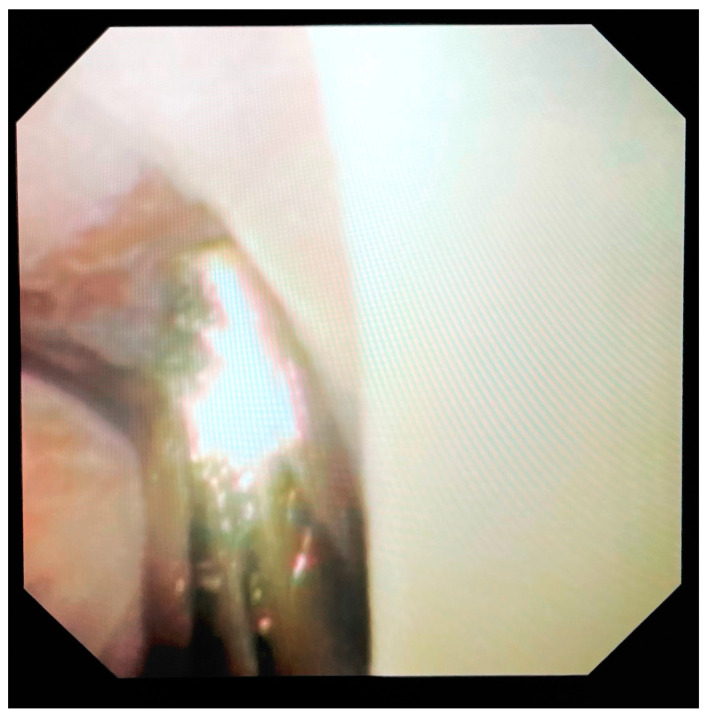
The probe of the thermometer in the proximity of the middle calyx papilla.

**Figure 3 medicina-61-01335-f003:**
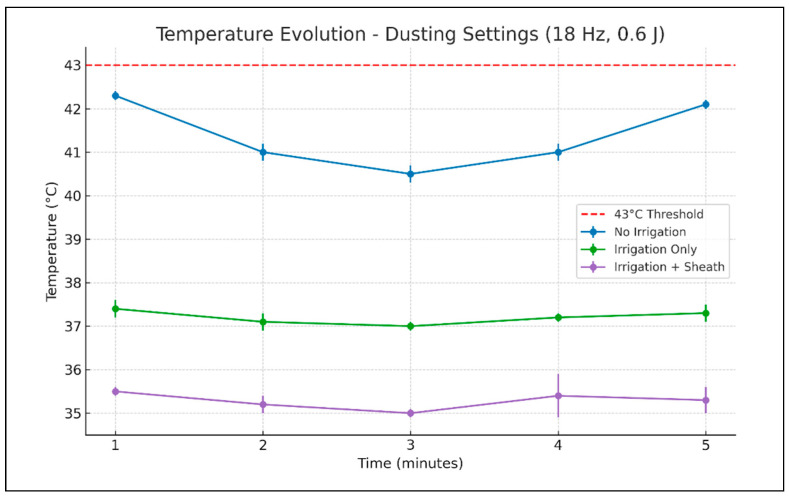
Evolution of temperature recorded in the middle calyx using dusting laser settings, during 5 min of continuous laser activation (mean values ± SD).

**Figure 4 medicina-61-01335-f004:**
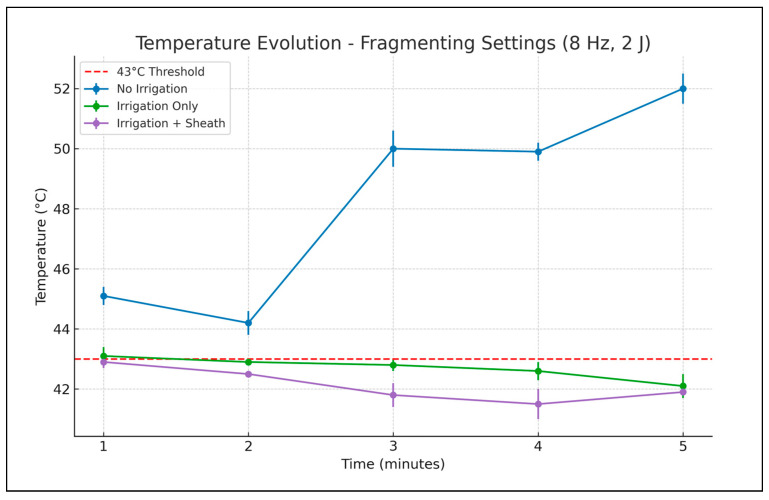
Evolution of temperature recorded in the middle calyx using fragmenting laser settings, during 5 min of continuous laser activation (mean values ± SD).

**Table 1 medicina-61-01335-t001:** Temperatures recorded in the middle calyx using dusting laser settings, during 5 min of continuous laser activation (mean values ± SD).

Laser Activation (Minutes)	Temperature at 18 Hz, 0.6 J-No Irrigation, No Ureteral Access Sheath (Degrees Celsius)	Temperature at 18 Hz, 0.6 J-Irrigation, No Ureteral Access Sheath (Degrees Celsius)	Temperature at 18 Hz, 0.6 J-Irrigation, 10/12F Ureteral Access Sheath (Degrees Celsius)	*p*-Values
1	42.3 ± 0.1	37.2 ± 0.1	35.0 ± 0.1	*p* < 0.0001
2	41.0 ± 0.2	37.0 ± 0.2	36.2 ± 0.2	*p* < 0.0001
3	39.7 ± 0.4	36.6 ± 0.1	35.6 ± 0.1	*p* < 0.0001
4	39.7 ± 0.3	36.2 ± 0.1	36.2 ± 0.5	*p* < 0.001
5	42.1 ± 0.1	37.3 ± 0.2	33.3 ± 0.3	*p* < 0.0001

**Table 2 medicina-61-01335-t002:** Temperatures recorded in the middle calyx using fragmenting laser settings, during 5 min of continuous laser activation.

Laser Activation (Minutes)	Temperature at 8 Hz, 2 J-No Irrigation, No Ureteral Access Sheath (Degrees Celsius)	Temperature at 8 Hz, 2 J-Irrigation, No Ureteral Access Sheath (Degrees Celsius)	Temperature at 8 Hz, 2 J-Irrigation, 10/12F Ureteral Access Sheath (Degrees Celsius)	*p*-Values
1	45.1 ± 0.3	43.1 ± 0.3	42.9 ± 0.2	*p* < 0.0001
2	44.2 ± 0.4	42.9 ± 0.1	42.5 ± 0.1	*p* = 0.0001
3	50.0 ± 0.6	42.8 ± 0.2	41.8 ± 0.4	*p* < 0.0001
4	49.9 ± 0.3	42.6 ± 0.3	41.5 ± 0.5	*p* < 0.0001
5	52.0 ± 0.5	42.1 ± 0.4	41.9 ± 0.1	*p* < 0.0001

## Data Availability

The original contributions presented in this study are included in the article. Further inquiries can be directed to the corresponding authors.

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
