# Peer review of "Temperature Increase During Flexible Ureteroscopic Approach with Holmium Laser Lithotripsy: How Much Should We Be Concerned?†"

_medicina, 2025, doi:10.3390/medicina61081335_

Round 1
Reviewer 1 Report
Comments and Suggestions for Authors
Dear author
Thanks for your submission. I read your manuscript and have some questions.
1- why did you choose 5 mins for laser activation? Or why did you evaluate temp. During 5 mins? Because as you revealed, in some circumstances temp. Increased minute by minute. Why did not you continue laser activation till reaching maximum temp.?
2- if you wanted to evaluate the effect of increasing temp. On kidney tissue, you should made a biopsy from kidney and evaluate it pathologically, so you can not comment on the effect of increasing temp. On kidney tissue. Please omit the sentences about this subject in your manuscript.
3- please mention the novelty of your study or bold the defferences of your study with the others.
4- page 7, first paragraph, second line," pop-dustingin", what is the meaning of this word? Is this spelling error?
Comments on the Quality of English LanguageNon
Author Response
Dear author, Thanks for your submission. I read your manuscript and have some questions.
We thank the reviewer for taking time for the detailed reading and constructive feedback.
Query 1: Why did you choose 5 mins for laser activation? Or why did you evaluate temp. During 5 mins? Because as you revealed, in some circumstances temp. Increased minute by minute. Why did not you continue laser activation till reaching maximum temp.?
Response to reviewer:
Thank you very much for your comment and question. As you very perceptively suspected, choosing the amount of time for laser application was indeed a difficult decision and was the result of extensive reflection. We chose a 5 minute laser activation period to simulate a realistic maximum continuous activation window encountered during lithotripsy segments, while maintaining safety and consistency in our measurements. In real clinical practice, laser activation is typically intermittent. Our goal was not to identify the absolute maximum possible temperature over prolonged activation, but rather to evaluate temperature evolution under plausible, controlled conditions, and compare how this is affected by irrigation and ureteral access sheath usage. We tried to clarify this point in the revised manuscript. The following paragraph was added in Discussion section: "The choice of a 5-minutes continuous laser activation period was intended to simulate a maximal yet plausible clinical scenario, allowing a standardized comparison across various scenarios. The continuous activation is not very common in real-life practice, and may be dangerous. In this regard, some authors reported increased morbidity associated with this style of laser lithotripsy, especially for high-power machines [5]. The current approach enabled us to observe temperature evolution under consistent and controlled circumstances, rather than to determine absolute peak temperatures under prolonged exposure."
We are aware that no choice is perfect, and because of that, in order to improve transparency and stimulate reflection, the following caution was inserted in the newly added limitations section: "Finally, continuous 5-minutes laser activation scenario, while chosen to ensure reproducibility and observe cumulative thermal effects, may not fully reflect the intermittent laser usage patterns typically observed in clinical practice."
Query 2: If you wanted to evaluate the effect of increasing temp. On kidney tissue, you should made a biopsy from kidney and evaluate it pathologically, so you can not comment on the effect of increasing temp. On kidney tissue. Please omit the sentences about this subject in your manuscript.
Response to reviewer:
We agree that definitive conclusions about tissue damage require histopathological evidence. We decided to evaluate this changes ex vivo because our primary goal was to design a reproducible and ethically responsible model to isolate the thermal effects of laser settings, irrigation, and access sheath use. While in vivo models provide greater realism, the ex vivo setup allowed controlled conditions with minimal variability, yet emulating at a reasonable scale the conditions encountered in real-life.
Although our manuscript discuss literature regarding thermal thresholds and potential biological effects, we revised relevant sentences to avoid any direct speculation about histological damage in the absence of pathological assessment. This limitation is also now explicitly acknowledged in the revised Discussion section.
We kept in the manuscripts some description of studies evaluating in vivo models, to draw a more complete picture of thermal dynamics associated with flexible ureteroscopic approach and to highlight the many variables involved when evaluating this issue. Another reason to include the discussion of the above mentioned studies was to allow a thorough comparison of the limitations but also of the benefits of ex vivo models (such as ours) with in vivo ones. Where this was the case, we added a cautionary phrase to remind the readers that no histopatological validation ware performed in our study.
The following text from Discussion section “...so such lithotripsy should be relatively safe in real life settings.” was replaced by “. suggesting that, under the studied conditions, temperature elevations are limited and unlikely to reach levels typically associated with thermal injury. However, in the absence of in vivo validation or histopathological assessment, no direct conclusions can be drawn regarding tissue damage.”
Also, the following statement was included in the limitations section added at the end of Discussion section: "This study has several limitations that should be acknowledged. First, it was conducted using an ex vivo porcine model, which does not fully replicate the physiological conditions of a perfused, living system. In particular, the absence of blood flow, immune response, and thermoregulatory mechanisms limits the extrapolation of our findings to in vivo settings. A further limitation is the absence of histopathological assessment, which was not pursued given the ex vivo nature of the model. Without vascular perfusion and tissue viability, such analysis would not yield meaningful insights into thermal injury, and definitive conclusions about tissue-level effects cannot be drawn. Therefore, our findings are restricted to thermal dynamics, and any biological implications remain speculative. While in vivo models provide greater realism, the ex vivo setup allowed controlled conditions with minimal variability."
Query 3: Please mention the novelty of your study or bold the defferences of your study with the others.
Response to the reviewer:
We thank the reviewer for this important observation. We have revised the Introduction and Discussion to better highlight the novelty of our study, particularly in: simulating common real-life laser settings and irrigation scenarios, providing direct temperature data under both dusting and fragmenting modes in a model built with animal tissues and evaluating the specific impact of access sheath use in a structured model.
The following paragraph was inserted at the end of Introduction section: “To our knowledge, this is one of the few studies to systematically evaluate temperature evolution under commonly used Holmium laser settings in a structured ex vivo model, while directly comparing the impact of irrigation and ureteral access sheath use. The strength of our work is that it measures this parameter on a biological model, simulating a plausible clinical scenario, in a controlled and standardized manner. This approach aims to provide practical insights relevant to real-life endourological procedures.”
Query 4: Page 7, first paragraph, second line," pop-dustingin", what is the meaning of this word? Is this spelling error?
Response to reviewer:
Indeed this was a typo and we apologize for it. The correct term is “pop-dusting,” which has been corrected in the manuscript. We also added between brackets a short explanation of the term to improve readability: "(achieving dusting using “pop-corn” effect, with short laser bursts of higher energy)”
Query 5: The English could be improved to more clearly express the research
Response to reviewer:
Thank you so much for the opportunity to improve our manuscript. Profesional language revision was performed. We corrected typos and grammatical errors and tried to rephrase the unnatural and awkward phrases to improve clarity and flow. All the other suggested linguistic improvements were also incorporated. The changes were highlighted in red in the manuscript.
Reviewer 2 Report
Comments and Suggestions for Authors
I have several concerns to the Authors:
1-Why they went through the ex-vivo model, rather than a live, anesthetized animal model, could be more physiological, as blood circulation impacts the thermal effect on the pelvicalyceal system.
2-As the authors utilized the Ex Vivo model, a histopathological assessment of renal tissues after thermal exposure could add to the study. However, the previous points can be mentioned as limitations of the study.
3- The introduction section needs improvement, considering the addition of references.
4- please, add the study limitations before the conclusion section.
Comments on the Quality of English Languagethe quality of English language needs improvement.
Author Response
I have several concerns to the Authors:
We thank the reviewer for investing time in evaluating our manuscript and for the constructive and valuable suggestions.
Query 1: Why they went through the ex-vivo model, rather than a live, anesthetized animal model, could be more physiological, as blood circulation impacts the thermal effect on the pelvicalyceal system.
Response to reviewer:
We acknowledge the reviewer’s point regarding physiological fidelity. Our primary goal was to design a reproducible and ethically responsible model to isolate the thermal effects of laser settings, irrigation, and access sheath use. While in vivo models provide greater realism, the ex vivo setup allowed controlled conditions with minimal variability, yet emulating at a reasonable scale the conditions encountered in real-life. We have now acknowledged this limitation in the Discussion section and also explained the rationale behind our choice.
Query 2: As the authors utilized the Ex Vivo model, a histopathological assessment of renal tissues after thermal exposure could add to the study. However, the previous points can be mentioned as limitations of the study.
Response to reviewer
We agree that histological evaluation is essential in assessing tissue-level effects; however, such analysis is meaningful in "in vivo" models, where vascular perfusion and inflammatory response are intact. In the absence of these physiological factors, we believe that histology from an ex vivo specimen would not yield relevant insights. As the reviewer constructively suggested, this rationale has been incorporated into the Discussion and listed as study limitations: "This study has several limitations that should be acknowledged. First, it was conducted using an ex vivo porcine model, which does not fully replicate the physiological conditions of a perfused, living system. In particular, the absence of blood flow, immune response, and thermoregulatory mechanisms limits the extrapolation of our findings to in vivo settings. A further limitation is the absence of histopathological assessment, which was not pursued given the ex vivo nature of the model. Without vascular perfusion and tissue viability, such analysis would not yield meaningful insights into thermal injury, and definitive conclusions about tissue-level effects cannot be drawn. Therefore, our findings are restricted to thermal dynamics, and any biological implications remain speculative. While in vivo models provide greater realism, the ex vivo setup allowed controlled conditions with minimal variability. "
Query 3: The introduction section needs improvement, considering the addition of references.
Response to reviewer
Thank you for this valuable suggestion and offering this opportunity for improvement of our manuscript. We have revised the Introduction to improve clarity and added references highlighting the clinical importance of temperature rise during laser lithotripsy.
The following paragraph was completely removed, as it was not relevant for our subject: The pyelocaliceal stones are the main indication of the method. Currently, there is widespread availability of advanced and enhanced flexible ureteroscopes (including the single use ones) allowing the urologists to constantly push the boundaries of this technique. Although the Holmium laser is still the most popular energy source for stone fragmentation, new laser types such as Thulium Fiber Laser (TFL) and Thulium:YAG are gaining recognition as valuable alternatives.
The phrase "In these circumstances the raise of temperature during laser lithotripsy is a popular topic.” was replaced by “The thermal effect generated during laser activation—especially at higher power settings or with prolonged firing—has raised increasing concern regarding its potential to injure renal tissues, thus becoming a popular study topic".
The paragraph "In the last years, numerous papers tried to evaluate the impact of this energy source when used during flexible ureteroscopy on the irrigation fluid temperature. It also tried to assess the effect of this phenomenon over the renal tissues. Immediate thermal tissue injury, as well as delayed damage (in this circumstances with the involvement of the healing process), were evaluated in several research studies. Nevertheless, despite the substantial evidence available on this subject, the risk and extent of such type of temperature induced harm remains a process that is not fully understood.” was replaced by “In the last years, numerous experimental studies tried to demonstrate that irrigation parameters, laser settings, and anatomical constraints significantly influence temperature dynamics within the pyelocaliceal system [1, 2]. A critical threshold of 43 °C is often cited as a marker of potential thermal injury, though the precise biological relevance of this value remains debated [3, 4]. In clinical settings, irrigation and the use of ureteral access sheaths may mitigate thermal rise, yet real-time temperature monitoring is rarely performed, and safety assumptions are largely extrapolated from preclinical data.”.
Also, the following paragraph was added at the end of Introduction section to highlight the novelty and importance of our study "To our knowledge, this is one of the few studies to systematically evaluate temperature evolution under commonly used Holmium laser settings in a structured ex vivo model, while directly comparing the impact of irrigation and ureteral access sheath use. The strength of our work is that it measures this parameter on a biological model, simulating a plausible clinical scenario, in a controlled and standardized manner. This approach aims to provide practical insights relevant to real-life endourological procedures.”.
Query 4: Please, add the study limitations before the conclusion section.
Response to reviewer:
We want to thank again the reviewer for the thoughtful suggestion. We have included a dedicated “Study limitations” section prior to the Conclusion section, summarising the constraints of our experimental design. We believe that this paragraph will significantly improve the value and transparency of our paper:
"Study limitations
This study has several limitations that should be acknowledged.
First, it was conducted using an ex vivo porcine model, which does not fully replicate the physiological conditions of a perfused, living system. In particular, the absence of blood flow, immune response, and thermoregulatory mechanisms limits the extrapolation of our findings to in vivo settings.
A further limitation is the absence of histopathological assessment, which was not pursued given the ex vivo nature of the model. Without vascular perfusion and tissue viability, such analysis would not yield meaningful insights into thermal injury, and definitive conclusions about tissue-level effects cannot be drawn. Therefore, our findings are restricted to thermal dynamics, and any biological implications remain speculative. While in vivo models provide greater realism, the ex vivo setup allowed controlled conditions with minimal variability.
Furthermore, the temperature of the model was not maintained at core body levels, which may limit the direct extrapolation of the results to in vivo conditions. These factors were necessary trade-offs to allow for controlled, reproducible comparisons between different irrigation and access sheath scenarios.
Certain parameters—such as intra-pyelocaliceal pressure and exact fiber-to-thermometer distances—were not directly measured or standardised, and irrigation flow was based on gravity-driven passive flow without active flow control. While these factors introduce some experimental variability, they were intentionally preserved to reflect the variability and realism of clinical practice. Our goal was not to create a fully mechanised setup, but rather to simulate operative scenarios in a reproducible yet pragmatically relevant manner.
Finally, continuous 5-minutes laser activation scenario, while chosen to ensure reproducibility and observe cumulative thermal effects, may not fully reflect the intermittent laser usage patterns typically observed in clinical practice."
Query 5: The English could be improved to more clearly express the research
Response to reviewer:
Thank you so much for the opportunity to improve our manuscript. Profesional language revision was performed. We corrected typos and grammatical errors and tried to rephrase the unnatural and awkward phrases to improve clarity and flow. All the other suggested linguistic improvements were also incorporated. The changes were highlighted in red in the manuscript.
Reviewer 3 Report
Comments and Suggestions for Authors
The manuscript is well structured and very clinically relevant. The subject of thermal injury is of critical importance during lithotripsy.
Howerver then lack of an in depth statistical analysis reduces the study scientific rigor.
Furthermore in endourology, operative situations in which continuous irrigation is not activated is rare.
The authors should address these two issues to achieve a more scientific rigor
Author Response
The manuscript is well structured and very clinically relevant. The subject of thermal injury is of critical importance during lithotripsy. The authors should address these two issues to achieve a more scientific rigor
We are grateful for the reviewer’s positive evaluation of the manuscript’s structure and clinical relevance, as well as for the insightful suggestions to enhance its scientific rigor.
Query 1: Howerver then lack of an in depth statistical analysis reduces the study scientific rigor.
Answer to reviewer:
We agree that statistical rigor is very important. The aim of our experimental design was to offer a descriptive comparison of temperature profiles across different commonly used laser settings and irrigation conditions and to assess how often dangerous temperatures are reached rather than to test a specific hypothesis using inferential statistics. Given the controlled, repeated-measure nature of the setup and the use of precise, continuous temperature recordings, we initially considered the tabulated temperature progression to be an appropriate and interpretable form of data presentation. However, we clearly understand your point, and after reflecting at your query, we understood that adding simple statistic study will add complexity to the study and offer essential informations to some readers.
So we performed the following actions:
- we added graphics (Figure 3 and 4) to present temperature variations (already described in tables) in a fashion that will increase readability and overall reader’s experience
- performed simple statistical analysis (one-way ANOVA and post hoc analysis); there were described in Material and Methods section, their results included in the Results section and further commented in Discuassion section.
- included in both tables and graphics the standard deviation values
- included in the tables the p value generated by the statistical analysis and commented it in the Discussion section
We thank you so much for the opportunity to improve our study.
Query 2: Furthermore in endourology, operative situations in which continuous irrigation is not activated is rare.
Response to reviewer:
We totally agree with the reviewer that continuous irrigation is standard in real-life endourological practice. Several previously published experimental studies (e.g., Rezakahn Khajeh et al., Okhunov et al.) have evaluated similar no irrigation scenarios. These indeed are probably never encountered in real life and temperature raise during these scenarios should never be a reason for concern during real interventions. We also interpreted this scenario with caution regarding its clinical applicability. We appreciate the reviewer’s perceptive observation and also for the opportunity to clarify our intent.
We underline that we decided to include such a scenario to serves as a reference/control condition, offering an upper-bound estimate of thermal risk in the absence of fluid-mediated cooling.
To address the reviewer’s concern and to avoid readers misunderstanding, we tried to clarify this rationale in the Discussion section, emphasising that this setting is not meant to reflect clinical practice but to provide a benchmark for interpreting the protective effects of irrigation and sheath use. It serves also as an preliminary step to include the direct argumentation of why this probably never encountered in real life.
The following paragraph was included in the beginning of the Discussion section: “Although the no-irrigation condition does not reflect typical operative endourological practice, it was included as a reference benchmark for comparison, consistent with prior experimental literature. This scenario allowed us to better illustrate the protective effects of irrigation and access sheath use in mitigating temperature rise during laser activation.”
Reviewer 4 Report
Comments and Suggestions for Authors
The idea of the article is interesting however it is hardly questionable that a " freshly
harvested porcine urinary system" might represent the reality of in vivo condition.
Moreover, it is described the "room temperature" but not the temperature of the model, therfore for sure this does not reflect in any way the real condition of a "heart beating" patient or animal model.
The idea and the concept of the paper is new and interesting but i do not think that the way the study is conducted might have any significant scientific relevance.
Author Response
Query 1: The idea of the article is interesting however it is hardly questionable that a " freshly
harvested porcine urinary system" might represent the reality of in vivo condition.
Response to reviewer:
We thank the reviewer for the thoughtful critique and recognition of the interest of our concept.
We agree that ex vivo models have inherent limitations in replicating the complete physiological environment of an in vivo system, especially regarding factors such as tissue perfusion, systemic thermoregulation, and metabolic activity. However, our study was explicitly designed to evaluate temperature variation under as much as possible controlled and reproducible conditions, focusing on the impact of irrigation and ureteral access sheath use—variables that can be ethically and practically isolated in ex vivo models but are harder to control in vivo.
This reality was acknowledged in the newly introduced Study limitations section, at the end of Discussion: "This study has several limitations that should be acknowledged. First, it was conducted using an ex vivo porcine model, which does not fully replicate the physiological conditions of a perfused, living system. In particular, the absence of blood flow, immune response, and thermoregulatory mechanisms limits the extrapolation of our findings to in vivo settings. A further limitation is the absence of histopathological assessment, which was not pursued given the ex vivo nature of the model. Without vascular perfusion and tissue viability, such analysis would not yield meaningful insights into thermal injury, and definitive conclusions about tissue-level effects cannot be drawn. Therefore, our findings are restricted to thermal dynamics, and any biological implications remain speculative. While in vivo models provide greater realism, the ex vivo setup allowed controlled conditions with minimal variability."
Query 2: Moreover, it is described the "room temperature" but not the temperature of the model, therfore for sure this does not reflect in any way the real condition of a "heart beating" patient or animal model.
Response to reviewer:
You are perfectly right and we truly appreciate this observation. In this regard we have now clarified in the Materials and Methods section the conditions in which the porcine system was used within 3 hours post-harvest, maintained at ambient conditions, and that no active heating was applied, thus representing a limitation which we have now also explicitly acknowledged in the Discussion section.
The following phrase was adde in the Material and Methods section: "The animal organs were used within 3 hours post-harvest, being maintained at ambient conditions; no measures of active heating were applied."
The following phrase was added to the Study limitations section: "Furthermore, the temperature of the model was not maintained at core body levels, which may limit the direct extrapolation of the results to in vivo conditions. These factors were necessary trade-offs to allow for controlled, reproducible comparisons between different irrigation and access sheath scenarios.”
Query 3: The idea and the concept of the paper is new and interesting but i do not think that the way the study is conducted might have any significant scientific relevance.
Response to reviewer:
We thank you again for acknowledging the novelty and interest in our study. We respectfully note that while this setup cannot perfectly simulate a “heart-beating” patient, numerous published studies (e.g., Okhunov et al., Aldoukhi et al.) have used ex vivo or in vitro models to explore thermal dynamics during laser lithotripsy. Our findings remain relevant in identifying trends and thresholds of temperature elevation under various irrigation scenarios, and we believe they contribute meaningful data to support safer clinical practice. Also, the setup for our model in different than other models and we are sure that it will help drawing a more complete picture on this issue. We also included simple statistical analysis to improve the evaluation. We hope that the reviewer will find the modifications to the manuscript valuable and adding depth, complexity and scientific rigor to it.
Reviewer 5 Report
Comments and Suggestions for Authors
Peer Review Report
- Summary
This manuscript explores temperature changes during flexible ureteroscopy using Holmium:YAG laser under different settings, using an ex vivo porcine model. They test two common laser settings (dusting and fragmenting) with and without irrigation and with or without ureteral access sheath.
The topic is relevant, specially for endourologists who work with high-power lasers routinely. The study confirms that proper irrigation and access sheath use can reduce thermal risks during lithotripsy, which is helpful in real-world scenarios. While the overall idea is good, the manuscript could benefit from more clear methodological reporting, some statistical analysis, and a better writting in general.
- General Concept Comments
The authors touch on an important safety concern in endourology, and the ex vivo model is designed well to simulate clinical settings. That being said, some limitations reduce the strenght of the conclusions:
- The paper lacks statistical analysis. The differences in temperature between conditions are meaningful but not formally compared.
- Key parameters like irrigation flow rate, intra-pyelocaliceal pressure, or fiber-to-probe distance were not standardized or even reported.
- The discussion is quite long, and sometimes repetitive. It could be more focused on practical implications for surgeons.
- The experimental design is simple and useful, but some clinical realism is lacking (e.g., 5 minutes continuous laser firing is not what we do in practice).
- The language needs polishing, some sentences are quite awkward or unclear.
The study provides useful insight, but it feels more like a pilot study or a technical note, not yet a full experimental paper.
- Article-Specific Comments
Introduction
- The background is generally informative but could be shortened. Too much space is spent on general FURS history and not enough on why temperature matters.
- Reference to 43ºC threshold is appropiate, although more recent studies have debated that point.
Methods
- Important details are missing: how was flow rate controlled? Was the laser fiber kept at a constant distance from the thermometer?
- No randomization or repetition is mentioned. Were the measures repeated to reduce variability?
- The use of continuous 5 minute laser seems unrealistic. Please justify this choice.
Results
- Tables are clear and show expected results: no irrigation = more heat.
- A graph would really help readers visualize the temp. curves over time.
- The lack of statistical analysis is a problem. Even some simple comparisons would strenghten the findings.
Discussion
- The authors discuss relevant literature and show that their findings are consistent with prior reports.
- However, too many references are used to say basically the same thing: irrigation cools, sheath helps.
- They do mention that exact thresholds for tissue damage are still debated, which is good.
- The part about cooling after laser stops is interesting and worth emphasizing more.
Conclusions
- The conclusion is reasonable. The final statement that laser lithotripsy is “reasonably safe” is acceptable, though quite vague.
- Writing Style and Errors
- Language is often awkward. For example:
- “Laser was indwelled directly” → unclear; should be “inserted”
- “Temperature become 20.3ºC” → typo, should be “became”
- “Evaluation was performed initially…” → better as “Initially, we evaluated…”
- Some phrases are unnatural: “it is presumed that” or “rendering the procedure safe” → consider more direct language
- Several minor typos and grammar mistakes throughout — a thorough language revision is needed
- Overall Recommendation
Major Revision
The study adds to the ongoing discussion on thermal safety in laser lithotripsy and provides helpful data from a realistic model. However, to be acceptable for publication, the authors should:
- Add at least basic statistical comparisons
- Clarify and improve the methods section
- Rewrite parts of the discussion to make it more concise and focused
- Improve the English language overall
With those changes, the manuscript would provide a valuable contribution to the field.
Author Response
Summary: This manuscript explores temperature changes during flexible ureteroscopy using Holmium:YAG laser under different settings, using an ex vivo porcine model. They test two common laser settings (dusting and fragmenting) with and without irrigation and with or without ureteral access sheath. The topic is relevant, specially for endourologists who work with high-power lasers routinely. The study confirms that proper irrigation and access sheath use can reduce thermal risks during lithotripsy, which is helpful in real-world scenarios. While the overall idea is good, the manuscript could benefit from more clear methodological reporting, some statistical analysis, and a better writting in general.
Response to reviewer:
We thank the reviewer for investing time in evaluating in so much detail our manuscript and also for the constructive suggestions. We truly believe that answering reviewers queries, the value and complexity of the manuscript significantly improved.
General Concept Comments: The authors touch on an important safety concern in endourology, and the ex vivo model is designed well to simulate clinical settings. The study provides useful insight, but it feels more like a pilot study or a technical note, not yet a full experimental paper. That being said, some limitations reduce the strenght of the conclusions:
Query 1: The paper lacks statistical analysis. The differences in temperature between conditions are meaningful but not formally compared.
Response to reviewer:
Thank you for acknowledging the importance of the topic of our study. Using the opportunity that your review is offering and following your suggestion, we added simple statistical analysis in the manuscript - this is detailed in the next paragraphs.
Query 2: Key parameters like irrigation flow rate, intra-pyelocaliceal pressure, or fiber-to-probe distance were not standardized or even reported.
Response to reviewer:
We thank you for highlighting this issue. We have now included additional methodological details in the Material and Methods section and also we explained our decisions in the Discussion section and newly introduced Study limitations section.
Query 3: The discussion is quite long, and sometimes repetitive. It could be more focused on practical implications for surgeons.
Response to reviewer:
Discussion section was reorganised in order to improve clarity and text flow.
Query 4: The experimental design is simple and useful, but some clinical realism is lacking (e.g., 5 minutes continuous laser firing is not what we do in practice).
Response to reviewer:
Thank you very much for your comment and question. As you very perceptively suspected, choosing the amount of time for laser application was indeed a difficult decision and was the result of extensive reflection. We chose a 5 minute laser activation period to simulate a realistic maximum continuous activation window encountered during lithotripsy segments, while maintaining safety and consistency in our measurements. In real clinical practice, laser activation is typically intermittent. Our goal was not to identify the absolute maximum possible temperature over prolonged activation, but rather to evaluate temperature evolution under plausible, controlled conditions, and compare how this is affected by irrigation and ureteral access sheath usage. We tried to clarify this point in the revised manuscript. The following paragraph was added in Discussion section: "The choice of a 5-minutes continuous laser activation period was intended to simulate a maximal yet plausible clinical scenario, allowing a standardized comparison across various scenarios. The continuous activation is not very common in real-life practice, and may be dangerous. In this regard, some authors reported increased morbidity associated with this style of laser lithotripsy, especially for high-power machines [5]. The current approach enabled us to observe temperature evolution under consistent and controlled circumstances, rather than to determine absolute peak temperatures under prolonged exposure."
We are aware that no choice is perfect, and because of that, in order to improve transparency and stimulate reflection, the following caution was inserted in the newly added limitations section: "Finally, continuous 5-minutes laser activation scenario, while chosen to ensure reproducibility and observe cumulative thermal effects, may not fully reflect the intermittent laser usage patterns typically observed in clinical practice."
Query 5: The language needs polishing, some sentences are quite awkward or unclear.
Response to reviewer:
The manuscript was professionally edited to solve this linguistic issues. Thank you again for the opportunity.
Article-Specific Comments
Query 6: Introduction: The background is generally informative but could be shortened. Too much space is spent on general FURS history and not enough on why temperature matters.
Response to reviewer:
Thank you for your constructive critique. The Introduction section was rewritten. General fURS history section was shortened and comments regarding why temperature matters were inserted, including references.
Query 7: Introduction: Reference to 43ºC threshold is appropiate, although more recent studies have debated that point.
Response to reviewer:
Thank you for your comment. We also consider this a very important discussion issue. Reference to 43 degrees threshold was introduced now also in the Introduction. The Discussion regarding it was strengthened.
Query 8: Methods: Important details are missing: how was flow rate controlled? Was the laser fiber kept at a constant distance from the thermometer?
Response to reviewer:
Thank you for your comments. These are definitely important issues to clarify. The irrigation flow was based on gravity-driven passive flow without any active control. Although this may be standard in many urological departments, it may be interpreted as a limitation in others. Fiber-to-thermometer distance was not standardized. It was manipulated as during a day by day operation, contributing to the real-life simulating scenario (as this was our main objective).
The following phrase was added to the Materials and Methods section: "The fiber-to-thermometer distance was not standardized, in an attempt to replicate real-life intraoperative conditions. For that, the laser fiber was manually maintained at an aproximately constant distance of about 2 mm from the thermometer tip under direct vision.”. The flow rate controll was described at the end of Materials and Methods section.
To improve transparency and readability, the following phrase was added to the Study limitations section: "Certain parameters—such as intra-pyelocaliceal pressure and exact fiber-to-thermometer distances—were not directly measured or standardised, and irrigation flow was based on gravity-driven passive flow without active flow control. While these factors introduce some experimental variability, they were intentionally preserved to reflect the variability and realism of clinical practice. Our goal was not to create a fully mechanised setup, but rather to simulate operative scenarios in a reproducible yet pragmatically relevant manner.”
Query 9: Methods: No randomization or repetition is mentioned. Were the measures repeated to reduce variability?
Response to reviewer:
We acknowledge that the description of the mathodology was not optimal. We regret these were not clearly reported. As we mentioned we performed sets of measurements (triplicates to be more specific). We added in the Material and Methods section a more thorough description of these determinations. Also, following your suggestion we included a simple statistical analysis, so we added more deteils in the tables such as standard deviation of our measurement.
Query 10: Methods: The use of continuous 5 minute laser seems unrealistic. Please justify this choice.
Thank you very much for your comment and question. As you very perceptively suspected, choosing the amount of time for laser application was indeed a difficult decision and was the result of extensive reflection. We chose a 5 minute laser activation period to simulate a realistic maximum continuous activation window encountered during lithotripsy segments, while maintaining safety and consistency in our measurements. In real clinical practice, laser activation is typically intermittent. Our goal was not to identify the absolute maximum possible temperature over prolonged activation, but rather to evaluate temperature evolution under plausible, controlled conditions, and compare how this is affected by irrigation and ureteral access sheath usage. We tried to clarify this point in the revised manuscript. The following paragraph was added in Discussion section: "The choice of a 5-minutes continuous laser activation period was intended to simulate a maximal yet plausible clinical scenario, allowing a standardized comparison across various scenarios. The continuous activation is not very common in real-life practice, and may be dangerous. In this regard, some authors reported increased morbidity associated with this style of laser lithotripsy, especially for high-power machines [5]. The current approach enabled us to observe temperature evolution under consistent and controlled circumstances, rather than to determine absolute peak temperatures under prolonged exposure."
We are aware that no choice is perfect, and because of that, in order to improve transparency and stimulate reflection, the following caution was inserted in the newly added limitations section: "Finally, continuous 5-minutes laser activation scenario, while chosen to ensure reproducibility and observe cumulative thermal effects, may not fully reflect the intermittent laser usage patterns typically observed in clinical practice."
Query 11: Results: Tables are clear and show expected results: no irrigation = more heat. A graph would really help readers visualize the temp. curves over time.
Response to reviewer:
Thank you so much for your valuable suggestion. We inserted two graphics (Figure 3 and Figure 4) describing temperature evolution using dusting and respectively fragmentation settings. We believe that they significantly improved readability and overall readers experience, and we thank you for that.
Query 12 Results: The lack of statistical analysis is a problem. Even some simple comparisons would strenghten the findings.
Response to reviewer:
We agree that statistical rigor is very important. The aim of our experimental design was to offer a descriptive comparison of temperature profiles across different commonly used laser settings and irrigation conditions and to assess how often dangerous temperatures are reached rather than to test a specific hypothesis using inferential statistics. Given the controlled, repeated-measure nature of the setup and the use of precise, continuous temperature recordings, we initially considered the tabulated temperature progression to be an appropriate and interpretable form of data presentation. However, we clearly understand your point, and after reflecting at your query, we understood that adding simple statistic study will add complexity to the study and offer essential informations to some readers.
So we performed the following actions:
- we added graphics (Figure 3 and 4) to present temperature variations (already described in tables) in a fashion that will increase readability and overall reader’s experience
- performed simple statistical analysis (one-way ANOVA and post hoc analysis); there were described in Material and Methods section, their results included in the Results section and further commented in Discuassion section.
- included in both tables and graphics the standard deviation values
- included in the tables the p value generated by the statistical analysis and commented it in the Discussion section
We thank you so much for the opportunity to improve our study.
Query 13: Discussion: The authors discuss relevant literature and show that their findings are consistent with prior reports. However, too many references are used to say basically the same thing: irrigation cools, sheath helps.
Response to reviewer:
Discussion section was reorganised in order to improve its clarity and text flow.
Query 14: Discussion: They do mention that exact thresholds for tissue damage are still debated, which is good.
Response to reviewer:
Thank you so much for your observation and for acknowledging this as one of the key issues. One of the main problems with intrarenal tempoerature studies is that this thresholds are frequently cited but still (also in our opinion) highly debatable and misused.
Query 15: Discussion: The part about cooling after laser stops is interesting and worth emphasizing more.
Response to reviewer:
Thank you for this valuable observation and for the opportunity to highlight a important practical point. We agree that the rapid temperature decline following laser deactivation, particularly with irrigation, is clinically meaningful. Although this was not a primary endpoint, we have expanded the Discussion section to emphasize this finding and its relevance for safe endourological practice. The following phrase was added:
"A notable finding in our study is the rather rapid temperature decline observed after laser deactivation, particularly under conditions with active irrigation. This sharp decrease highlights the role of irrigation not only in mitigating temperature rise during lithotripsy but also in facilitating fast thermal recovery once energy delivery ceases. Although not a primary endpoint, this observation reinforces the clinical benefit of using continuous irrigation and intermittent laser activation to limit tissue exposure to prolonged elevated temperatures."
Query 16: Conclusions: The conclusion is reasonable. The final statement that laser lithotripsy is “reasonably safe” is acceptable, though quite vague.
Response to reviewer:
Thank you for this thoughtful comment. We agree that the phrase “reasonably safe” may benefit from clarification. To better reflect the findings and clinical implications, we have slightly revised the second phrase of the Conclusion section.
The phrase "However, in closer to real life settings, the magnitude of this phenomenon is limited, rendering the flexible ureteroscopic approach with laser lithotripsy a reasonably safe procedure.” was replaced with "Our results suggest that, in most real-life settings, laser lithotripsy can be performed safely when appropriate thermal protective measures (such as continuous irrigation and laser activation in time-limited bursts) are applied to prevent excessive intrarenal temperature rise."
Query 17: Writing Style and Errors: Language is often awkward. For example:Laser was indwelled directly” → unclear; should be “inserted”; "Temperature become 20.3ºC” → typo, should be “became”; "Evaluation was performed initially…” → better as “Initially, we evaluated…”. Some phrases are unnatural: “it is presumed that” or “rendering the procedure safe” → consider more direct language. Several minor typos and grammar mistakes throughout — a thorough language revision is needed
Response to reviewer:
Thank you so much for your attention to details and for the opportunity to improve our manuscript. Profesional language revision was performed. We corrected typos and grammatical errors and tried to rephrase the unnatural and awkward phrases to improve clarity and flow. All the other suggested linguistic improvements were also incorporated. The changes were highlighted in red in the manuscript.
Overall Recommendation: Major Revision
The study adds to the ongoing discussion on thermal safety in laser lithotripsy and provides helpful data from a realistic model. However, to be acceptable for publication, the authors should:
- Add at least basic statistical comparisons
- Clarify and improve the methods section
- Rewrite parts of the discussion to make it more concise and focused
- Improve the English language overall
With those changes, the manuscript would provide a valuable contribution to the field.
Response to reviewer:
We are especially grateful for the thoughtful tone of this review, which has significantly improved both the clarity and scientific rigor of our manuscript. We believe the revised version now better reflects the value of the findings while acknowledging its technical scope and limitations. To summarize, we added simple statistical analysis and commented on the results in the Discussion section, clarified and detailed the Materials and Methods section, Reorganized and rewritten the Discussion section for more clarity and flow, profesionally edited the text for English language overall. We sincerely thank the reviewer for this detailed, insightful, and constructive critique and hope that we answered to all queries in a satisfactory manner.
Round 2
Reviewer 3 Report
Comments and Suggestions for Authors
The manuscript presents a structured ex vivo study on temperature changes during Holmium laser lithotripsy under different conditions. The study design is adequate, and the data are clearly presented. Some aspects could be better detailed (for example, the irrigation flow rate, thermometer specifications, and the standardization of the laser fiber distance from the probe). A brief discussion of effect size and the clinical impact of temperature differences would also be useful.
Author Response
1. The manuscript presents a structured ex vivo study on temperature changes during Holmium laser lithotripsy under different conditions. The study design is adequate, and the data are clearly presented.
Response to reviewer:
We sincerely thank the reviewer for their thoughtful evaluation and for taking the time to analyze our manuscript in detail. We are grateful for the recognition of the study design and the clarity of the data presentation, and we have answered the additional queries below.
2. Some aspects could be better detailed:
- the irrigation flow rate
Response to reviewer:
Thank you for this valuable suggestion. We have detailed informations regarding gravitational irrigation in the Methods section and provided a mean value of the measured flow rate. The Phrase "We used hydrostatic pressure, with the saline bags being suspended at 1 m above the kidneys.” was modified as follows: "We used hydrostatic pressure, with the saline bags being suspended at 100 cm above the kidneys, providing a flow of approximately 35 mL/min.”
- thermometer specifications
Response to reviewer:
We agree this is an important technical detail to be added in the manuscript.
The following phrase was added to the revised Materials and Methods section: "Temperature measurements were performed using a TP-300 digital thermometer equipped with a K-type thermocouple probe, with an operating range of –50°C to +300°C and a resolution of 0.1°C."
- the standardization of the laser fiber distance from the probe
Response to reviewer:
Thank you for your constructive comment. We agree that precise standardization of the fiber-to-thermometer distance can influence temperature measurements. However, we chose to maintain a realistic setup to simulate intraoperative conditions. The laser fiber was manually held at an approximately constant distance of ~2 mm from the thermometer tip under direct visual control. We explained this methodological choice in the revised Materials and Methods section, and also its implications in the Study Limitations paragraph.
This phrase was added in the Materials and Methods section: The fiber-to-thermometer distance was not standardized, in an attempt to replicate real-life intraoperative conditions. For that, the laser fiber was manually maintained at an aproximately constant distance of about 2 mm from the thermometer tip under direct vision.
This phrase was added in the Discussion section: Certain parameters—such as intra-pyelocaliceal pressure and exact fiber-to-thermometer distances—were not directly measured or standardised, and irrigation flow was based on gravity-driven passive flow without active flow control. While these factors introduce some experimental variability, they were intentionally preserved to reflect the variability and realism of clinical practice. Our goal was not to create a fully mechanised setup, but rather to simulate operative scenarios in a reproducible yet pragmatically relevant manner.
3. A brief discussion of effect size and the clinical impact of temperature differences would also be useful.
Response to reviewer:
Thank you for this thoughtful suggestion. While effect size is not expressed numerically due to the descriptive nature of our experimental setup, we have added a concise statement to the Discussion emphasizing the magnitude and clinical significance of the observed temperature differences.
We believe that the recorded temperature differences between experimental conditions may be clinically relevant when considering thermal safety during laser lithotripsy. As noted, while temperatures above 43°C are often cited as potentially harmful, tissue injury is not determined by a single threshold but rather by the combination of peak temperature and duration of exposure. Thus, reducing intrarenal temperature even by a few degrees, may lower the risk of cumulative thermal injury during prolonged activation periods. While we did not calculate formal effect size metrics, the significant ANOVA results across multiple settings suggest that experimental conditions (and especially irrigation and UAS) may have a substantial impact on temperature outcomes.